# Design and Synthesis of Immunoadjuvant QS-21 Analogs and Their Biological Evaluation

**DOI:** 10.3390/biomedicines12020469

**Published:** 2024-02-19

**Authors:** Wei Yuan, Ziming Wang, Yening Zou, Guojun Zheng

**Affiliations:** 1State Key Laboratories of Chemical Resources Engineering, Beijing University of Chemical Technology, Beijing 100029, China; 15003331983@163.com (W.Y.); 17812118992@163.com (Z.W.); 2Sinovac Life Sciences Co., Ltd., Beijing 102601, China

**Keywords:** analog, hemolytic, QS-21, synthesis

## Abstract

A series of novel immunoadjuvant QS-21 analogs were synthesized, and their effects on the in vitro hemolysis of red blood cells were evaluated using QS-21 as a control and hemolytic properties as an index. Our results show that all the QS-21 analogs had lower hemolytic effects than QS-21, and their concentrations exhibited a certain quantitative effect relationship with the hemolysis rate. Notably, saponin compounds **L1**–**L8** produced minimal hemolysis and showed lower hemolytic effects, warranting further investigation.

## 1. Introduction

The theoretical basis for immune adjuvants emerged more than forty years after the introduction of vaccines. They are defined as a class of compounds that induce the differentiation of the organism to thymus-derived helper T cell type 1 or 2 (Th1 or Th2) and significantly enhance the immune response to antigens. After years of intensive research and development, immune adjuvants have now become the primary solution for improving the immune response to vaccines [1], and their main function is to assist in the rapid production of antigens and enhance the immunogenicity of vaccines [2].

QS-21 (Figure 1), a class of natural saponin products with significant immunostimulatory effects, has attracted considerable interest due to its powerful adjuvant potency and has been used in several clinical trials in combination with various vaccines as an adjuvant [3]. Saponin QS-21 is the 21st component of the bark extract of Quillaja-saponaria Molina from South America, and it consists of four structural units: saponate glucoside elements [4], branched trisaccharides, bridged linear tetrasaccharides, and pseudodimeric acyl side chains [5]. It has been shown to induce not only antibody-based humoral immune responses (Th2) [6,7] but also to stimulate cellular immune responses (Th1) [8,9]. Currently, more than 100 vaccines containing QS-21 are in clinical studies [10], and vaccine indications include cancer [11], neurodegenerative diseases such as Alzheimer’s disease [12], as well as malaria [13], tuberculosis [14], hepatitis [15], and so on. However, their use in humans remains limited due to several limitations, including structural complexity, scarcity [16], chemical instability [17], severe immune side effects, expensive and restricted intellectual property procurement, dose-limiting toxicity [18], and adverse hemolytic effects [19,20]. Moreover, samples are difficult to obtain, and their mechanism of action remains to be elucidated. 

Several research groups have previously synthesized a series of QS-21 analogs and investigated the role of various fragments of QS-21 on their activity. Zeng et al. found a scheme characterized by Yu glycosylation to construct the challenging 3-O-glycosidic bonds of C23-oxooleanane triterpenoids. This provided a solution to the long-standing obstacle limiting the easy availability of the 3-O-glycosides of C23-oxooleanane triterpenoids, including QS-21 [21]. Roberto Fuentes et al. primarily studied the effect of terminal disaccharide modification on saponin conformer-related adjuvant activity, and combined with immunological results, it was found that the original rhamnose–xylose part within this domain was important for adjuvant activity [22]. The superiority of the echinocystic acid variants proposed by Mattia Ghirardello et al. makes them leading scaffolds for future mechanistic studies and synthetic vaccines based on saponin adjuvants [23].

In this study, four structural domains of QS-21 (i.e., fragment A—triterpene unit, fragment B—trisaccharide unit, fragment C—tetrasaccharide unit, and fragment D—acyl side chain) were modified and optimized, and 18 saponin analogs were individually designed and synthesized (Figure 2). The modification was included as follows: (1) fragment A: replacement of the quillaic acid with oleanolic acid, and introduction of the carbonyl and oxime functional groups at the C3 position; (2) fragments B and C: replacement of the original saccharide of QS-21 with inexpensive sugar, such as glucose, cellobiose, or maltotriose, and an attempt to remove the tetrasaccharide fragment; (3) fragment D: modification of the complex structure of the multi-chiral central acyl chain by introducing glucose, cellobiose, and maltotriose sugar at the end of the acyl chain and replacing the ester with an amide structure to improve the stability. Finally, 18 analogs were designed (Figure 2).

## 2. Materials and Methods

### 2.1. Chemical Synthesis

#### 2.1.1. General Considerations

All chemicals utilized for this research were of high quality and commercially available without further purification, unless otherwise stated. QS-21 was purchased from China Shanghai Yuanye Bio-Technology Co., Ltd. (Shanghai, China). Oleanolic acid was purchased from China Bide Pharmatech Co., Ltd. (Shanghai, China). Commercially available anhydrous dichloromethane (DCM), toluene, N, N-Dimethylformamide (DMF), methanol (MeOH), and ethanol (EtOH) were used to perform the reactions, unless otherwise stated. When necessitated, tetrahydrofuran (THF) was obtained through distillation over sodium/benzophenone.

High-resolution mass spectra were obtained on a Waters Xevo G2 Qtof mass spectrometer in the ESI mode. The ^1^H, ^13^C NMR spectra were determined on Bruker A V ANCE III HD 400 instruments using tetramethylsilaneas as an internal reference. The chemical shifts reported for the proton and carbon spectra were standardized to the particular NMR solvent utilized, and for this reason, the chemical shifts of the solvent are not detailed within each experimental procedure. The data are presented as follows: chemical shift, multiplicity (s = singlet, d = doublet, t = triplet, q = quadruplet, m = multiplet), J = coupling constant in hertz (Hz). 

#### 2.1.2. General Procedure to Prepare QS-21 Analogues **L1**–**L15**

The general procedure to prepare QS-21 analogues **L1**–**L15** is as follows. Taking the synthesis of QS-21 analogue **L2** as an example, oleanolic acid and its derivatives (1.00 eq) were dissolved in dry DCM (3 mL), oxalyl chloride (10.13 eq) was then added slowly dropwise at room temperature and stirred for 4 h. The reaction solution was evaporated, dissolved in toluene, and then evaporated. This process was repeated three times to remove the solvent. The above solid product was dissolved in anhydrous DCM solution (3 mL), amine material **D14** (1.00 eq) and 5 drops of triethylamine were added, and the solution was reacted at room temperature for 4 h. After the reaction, the reaction solution was concentrated under reduced pressure, evaporated to remove the solvent, then dissolved with DCM and extracted with water several times, washed with saturated NaCl, dried with anhydrous Na_2_SO_4_ dry, concentrated under vacuum, and purified by column chromatography to form **F2**. The product **F2** (1.00 eq) was dissolved in dry DCM/MeOH (1:2) (3 mL), added to freshly prepared sodium methoxide (1 mol/L) solution, stirred at room temperature for 2 h, then added to acidic cation exchange resin to neutral pH = 7, extracted, concentrated, and purified by column chromatography to give target compound **L2**. 


*Oleanolic acid-28-oyl-pentanol (**L1**)*


Oily solid, yield: 78.08%. ^1^H NMR (400 MHz, CDCl_3_) δ 7.15 (t, J = 6.0 Hz, 1H), 5.94 (t, J = 5.4 Hz, 1H), 5.35 (s, 1H), 4.61 (dd, J = 11.8, 4.2 Hz, 1H), 3.61 (td, J = 6.4, 3.2 Hz, 4H), 3.35–3.31 (m, 2H), 3.06–2.92 (m, 1H), 2.48 (d, 1H), 2.14 (d, 1H), 1.93–1.88 (m, 2H), 1.72 (d, 2H), 1.62–1.54 (m, 9H), 1.38–1.29 (m, 14H), 1.14 (s, 3H), 1.04 (d, 2H), 0.94 (d, 6H), 0.88 (d, 9H), 0.75 (s, 3H).^13^C NMR (101 MHz, CDCl_3_) δ 178.15, 145.00, 122.41, 84.50, 77.69, 77.68–75.89 (m), 76.32–75.89 (m), 62.42, 55.01, 53.32, 47.34, 46.65, 46.14, 42.07, 39.26, 37.89, 36.74, 34.02, 34.01–31.53 (m), 30.62, 29.10, 27.88, 27.16, 25.82, 25.60, 23.97–22.45 (m), 18.04, 16.66, 15.32. HRMS (ESI): [M + H]^+^ (C_35_H_59_NO_3_) calculated: 542.4568, found: 542.4564.


*Oleanolic acid-28-oyl-((5-(β-D-glucose)oxy)pentane) (**L2**)*


White solid, yield: 70.74%. ^1^H NMR (400 MHz, CDCl_3_) δ 6.10 (s, 1H), 5.37 (s, 1H), 4.94 (s, 1H), 4.78 (s, 1H), 4.47 (s, 1H), 4.29 (d, J = 7.6 Hz, 1H), 3.85 (d, J = 16.1 Hz, 3H), 3.54 (d, J = 9.5 Hz, 2H), 3.40–3.28 (m, 3H), 3.20 (dd, 1H), 3.02–2.91 (m, 1H), 2.49 (d, 2H), 1.99–1.87 (m, 3H), 1.78–1.68 (m, 2H), 1.66–1.43 (m, 14H), 1.42–1.17 (m, 8H), 1.15 (s, 3H), 0.98 (s, 3H), 0.91 (s, 3H), 0.89 (s, 6H), 0.78 (s, 3H), 0.74 (s, 3H). ^13^C NMR (101 MHz, CDCl_3_) δ 78.89, 77.32, 77.00, 76.68, 69.87, 55.08, 47.50, 46.93–46.73 (m), 46.23, 42.39–42.19 (m), 42.05, 39.34, 38.75, 38.45, 36.95, 34.12, 32.97, 30.70, 29.05, 28.11, 27.24, 27.01, 25.73, 23.52, 18.29, 16.94, 15.64, 15.42. HRMS (ESI): [M + H]^+^ (C_41_H_69_NO_8_) calculated: 704.5096, found: 704.5087.


*Oleanolic acid-28-oyl-((5-(β-D-cellobiose)oxy)pentane) (**L3**)*


White solid, yield: 61.3%. ^1^H NMR (400 MHz, CDCl_3_) δ 6.17 (s, 1H), 5.38 (s, 2H), 4.52 (s, 1H), 4.29 (s, 1H), 3.84 (s, 4H), 3.66 (s, 3H), 3.43 (s, 3H), 3.20 (s, 1H), 2.99 (s, 1H), 2.52 (s, 1H), 1.92 (s, 4H), 1.59 (d, J = 22.9 Hz, 13H), 1.44–1.30 (m, 5H), 1.26 (s, 3H), 1.22 (s, 2H), 1.15 (s, 3H), 1.01 (s, 2H), 0.98 (s, 3H), 0.90 (s, 9H), 0.77 (d, 6H).^13^C NMR (101 MHz, CDCl_3_) δ 171.15, 157.95, 122.36, 102.67, 101.77, 83.82, 79.51, 79.25, 78.82, 76.11, 75.79, 74.90, 73.02, 71.28, 69.94–69.66 (m), 69.33, 60.38, 50.80, 46.19, 43.96, 42.01, 39.36, 38.76, 36.97, 33.04, 30.71, 29.21, 28.19, 25.77, 23.71, 21.03, 17.02, 15.78, 15.47, 14.17. HRMS (ESI): [M + H]^+^ (C_47_H_79_NO_13_) calculated: 866.5625, found: 866.5633.


*Oleanolic acid-28-oyl-(5-((β-D-maltotriose)oxy)pentane) (**L4**)*


White solid, yield: 83.9%. ^1^H NMR (400 MHz, CDCl_3_)δ 8.35 (s, 1H), 7.24 (d, J = 7.3 Hz, 1H), 5.20 (s, 1H), 5.04–4.96 (m, 3H), 3.68–3.59 (m, 5H), 3.46 (d, J = 6.0 Hz, 2H), 3.40–3.29 (m, 6H), 3.25 (dd, J = 9.6, 3.7 Hz, 2H), 2.98 (ddd, 6H), 2.77 (d, J = 12.7 Hz, 1H), 1.78 (s, 1H), 1.70–1.32 (m, 17H), 1.31–1.13 (m, 8H), 1.07 (s, 3H), 0.86 (dd, J = 12.4, 5.9 Hz, 12H), 0.66 (d, J = 1.6 Hz, 6H). ^13^C NMR (101 MHz, DMSO) δ 158.76, 115.78, 79.72, 76.69, 73.43, 70.01, 60.97, 47.52, 46.24–45.08 (m), 41.32, 41.15–39.78 (m), 39.92, 39.92, 39.78, 39.90–39.21 (m), 39.08, 38.87, 32.93, 30.73, 29.07, 28.30, 27.01, 23.09, 22.49. HRMS (ESI): [M + H]^+^ (C_53_H_89_NO_18_) calculated: 1028.6153, found: 1028.6102.


*Oleanolic acid-28-oyl-hexanoate (**L5**)*


Oily solid, yield: 53.95%. ^1^H NMR (400 MHz, CDCl_3_) δ 5.93 (t, J = 5.4 Hz, 1H), 5.37 (t, J = 3.3 Hz, 1H), 3.66 (s, 3H), 3.36 (dd, J = 13.7, 6.5 Hz, 1H), 3.21 (dd, J = 11.1, 4.6 Hz, 1H), 2.99 (dd, J = 13.2, 4.8 Hz, 1H), 2.53–2.45 (m, 1H), 2.31 (t, J = 7.4 Hz, 2H), 1.91 (dd, J = 8.6, 3.2 Hz, 2H), 1.74 (d, J = 13.6 Hz, 4H), 1.61 (ddd, J = 17.3, 11.3, 6.1 Hz, 7H), 1.56–1.46 (m, 6H), 1.31 (dd, J = 23.1, 16.4 Hz, 8H), 1.16 (s, 3H), 0.99 (s, 3H), 0.91 (s, 3H), 0.90 (s, 6H), 0.78 (s, 3H), 0.75 (s, 3H).^13^C NMR (101 MHz, CDCl_3_) δ 173.98, 145.12, 122.66, 78.93, 77.32, 77.00, 76.68, 217.92–38.36 (m), 53.28, 72.35–46.92 (m), 50.22, 72.35–42.41 (m), 72.35–42.18 (m), 72.35–39.44 (m), 72.35–38.87 (m), 36.94, 34.12, 33.88, 32.97, 32.41, 30.71, 29.04, 28.07, 27.20, 26.57, 25.71, 24.52, 23.77, 23.54, 18.26, 16.89, 15.54, 15.33. HRMS (ESI): [M + H]^+^ (C_37_H_61_NO_4_) calculated: 584.4674, found: 584.4686.


*Oleanolic acid-28-oyl-(N-(5-hydroxypentyl)hexanamide) (**L6**)*


Oily solid, yield: 76.5%. ^1^H NMR (400 MHz, CDCl_3_) δ 5.33 (d, J = 14.4 Hz, 1H), 4.64 (s, 1H), 3.10 (q, J = 7.2 Hz, 6H), 2.82 (d, J = 9.7 Hz, 1H), 2.08–1.50 (m, 21H), 1.40 (t, J = 7.3 Hz, 12H), 1.25 (s, 3H), 1.14 (s, 3H), 0.92 (dd, J = 13.1, 6.0 Hz, 12H), 0.79 (d, 3H).^13^C NMR (101 MHz, CDCl_3_) δ 174.26, 168.23, 154.68, 134.02, 127.41, 104.98, 77.32, 77.00, 76.68, 69.78, 62.26, 54.79, 55.28, 48.34, 47.24, 45.72, 45.01, 37.95, 33.08, 32.48, 29.30, 23.69–23.60 (m), 18.90, 17.14, 16.61, 10.22, 8.60. HRMS (ESI): [M + H]^+^ (C_41_H_70_N_2_O_4_) calculated: 655.5409, found: 655.5381.


*Oleanolic acid-28-oyl- (N-(5-((β-D-glucose)oxy)pentyl)hexanamide) (**L7**)*


White solid, yield: 47.74%. ^1^H NMR (400 MHz, CDCl_3_) δ 6.89 (s, 1H), 6.14 (s, 1H), 5.37 (s, 1H), 4.28 (s, 1H), 3.81 (s, 4H), 3.56 (s, 2H), 3.46 (s, 1H), 3.34 (s, 1H), 3.29 (s, 2H), 3.18 (s, 2H), 2.98 (s, 1H), 2.50 (s, 1H), 2.18 (d, 1H), 1.87 (s, 1H), 1.57 (s, 12H), 1.46 (s, 7H), 1.30 (s, 6H), 1.25 (t, J = 7.1 Hz, 5H), 1.15 (s, 3H), 0.97 (s, 3H), 0.89 (s, 9H), 0.76 (s, 3H), 0.73 (s, 3H). ^13^C NMR (101 MHz, CDCl_3_) δ 171.12, 144.80, 102.53, 77.32, 77.00, 76.68, 60.36, 46.19, 42.06, 39.37, 38.74, 36.93, 32.98, 30.68, 29.04, 28.15, 26.67, 25.74, 25.34, 23.63, 23.19, 21.01, 16.98, 15.73, 15.41, 14.16. HRMS (ESI): [M + H]^+^ (C_47_H_80_N_2_O_9_) calculated: 817.5937, found: 817.5966. 


*Oleanolic acid-28-oyl-(N-(5-((β-D-cellobiose)oxy)pentyl)hexanamide) (**L8**)*


White solid, yield: 64.9%. ^1^H NMR (400 MHz, CDCl_3_) δ 7.97 (s, 1H), 7.94 (s, 1H), 6.06 (s, 1H), 5.74 (s, 1H), 5.35 (d, J = 18.8 Hz, 2H), 4.79 (s, 1H), 4.50 (s, 1H), 4.17 (s, 1H), 4.02 (d, J = 33.6 Hz, 2H), 3.80 (d, J = 38.5 Hz, 3H), 3.30 (s, 3H), 2.94 (d, J = 45.3 Hz, 6H), 2.49 (d, J = 11.4 Hz, 1H), 2.11 (s, 2H), 1.92 (s, 2H), 1.66–1.39 (m, 20H), 1.37 (s, 11H), 1.16 (s, 3H), 0.98 (s, 3H), 0.89 (d, J = 4.4 Hz, 9H), 0.76 (d, J = 11.6 Hz, 6H). ^13^C NMR (101 MHz, DMSO) δ177.40, 175.40, 152.33, 125.37, 117.88, 117.67, 93.38, 91.18, 87.93, 85.65, 85.46, 79.59, 77.85, 75.48, 71.45, 69.98, 68.30, 60.89, 53.87, 49.53, 47.33, 39.92–39.08 (m), 33.82, 32.36, 28.85, 22.61, 22.25, 15.89, 14.57, 8.56, 5.92. HRMS (ESI): [M + H]^+^ (C_53_H_90_N_2_O_14_) calculated: 979.6465, found: 979.6489.


*3-Oxo-oleanolic acid-28-oyl-pentyl alcohol (**L9**)*


Oily solid, yield: 71.91%. ^1^H NMR (400 MHz, CDCl_3_) δ 5.94 (t, J = 5.3 Hz, 1H), 5.37 (s, 1H), 3.61 (t, J = 6.2 Hz, 2H), 3.34 (dd, J = 13.3, 6.8 Hz, 1H), 3.00 (dd, J = 12.7, 5.5 Hz, 1H), 2.59–2.46 (m, 2H), 2.34 (ddd, J = 15.8, 6.3, 3.4 Hz, 1H), 2.03–1.81 (m, 5H), 1.78–1.23 (m, 21H), 1.14 (s, 3H), 1.06–1.00 (m, 9H), 0.88 (s, 6H), 0.79 (s, 3H).^13^C NMR (101 MHz, CDCl_3_) δ 178.20, 145.08, 122.38, 77.32, 77.00, 76.68, 62.49, 55.16, 47.39, 46.70, 46.24, 42.23, 39.32, 39.18, 36.63, 34.06, 32.91, 32.30, 32.02–31.90 (m), 31.84, 30.66, 29.14, 27.22, 26.37, 25.57, 23.70, 23.32, 21.42, 19.48, 16.77, 15.02. HRMS (ESI): [M + H]^+^ (C_35_H_57_NO_3_) calculated: 540.4412, found: 540.4423.


*3-Oxo-oleanolic acid-28-oyl-(5-((β-D-glucose)oxy)pentane) (**L10**)*


White solid, yield: 77.93%. ^1^H NMR (400 MHz, CDCl_3_) δ 6.10 (d, J = 5.0 Hz, 1H), 5.38 (s, 1H), 4.28 (d, J = 7.4 Hz, 1H), 3.84 (d, J = 14.1 Hz, 3H), 3.60–3.48 (m, 3H), 3.39–3.24 (m, 3H), 3.01–2.90 (m, 1H), 2.57–2.47 (m, 2H), 2.36 (d, J = 3.9 Hz, 1H), 1.92 (dd, J = 29.9, 15.8 Hz, 4H), 1.72–1.16 (m, 22H), 1.14 (s, 3H), 1.08–1.02 (m, 9H), 0.88 (s, 6H), 0.79 (s, 3H). ^13^C NMR (101 MHz, CDCl_3_) δ 178.55, 160.10, 144.92, 122.52, 102.82, 77.32, 77.00, 76.68, 75.60, 73.41, 69.85, 61.77, 55.16, 47.39, 47.05–46.01 (m), 42.17, 39.39, 39.10, 36.66, 34.08, 32.71, 31.85, 30.67, 29.09, 27.23, 26.43, 25.61, 23.44, 21.47, 19.52, 16.80, 15.10. HRMS (ESI): [M + H]^+^ (C_41_H_67_NO_8_) calculated: 702.4940, found: 702.4936.


*3-Oxo-oleanolic acid-28-oyl-(5-((β-D-cellobiose)oxy)pentane) (**L11**)*


White solid, yield: 74.40%. ^1^H NMR (400 MHz, CDCl_3_) δ 6.20 (s, 1H), 5.39 (s, 1H), 4.51 (s, 2H), 4.28 (s, 2H), 3.87 (d, J = 32.9 Hz, 6H), 3.69 (s, 4H), 3.31 (s, 2H), 2.99 (s, 1H), 2.53 (s, 2H), 2.36 (d, J = 13.7 Hz, 1H), 1.91 (d, J = 32.8 Hz, 5H), 1.44 (dd, J = 87.0, 35.7 Hz, 24H), 1.15 (s, 3H), 1.07 (s, 3H), 1.03 (s, 6H), 0.89 (s, 6H), 0.79 (s, 3H). ^13^C NMR (101 MHz, CDCl_3_) δ 102.67, 77.32, 77.00, 76.68, 73.02, 69.94–69.66 (m), 60.38, 46.19, 42.01, 39.36, 38.76, 36.97, 33.03, 32.44, 30.71, 29.21, 28.19, 25.77, 23.66, 21.03, 18.40, 17.02, 15.59, 14.17. HRMS (ESI): [M + H]^+^ (C_47_H_77_NO_13_) calculated: 864.5468, found: 864.5471.


*3-Oxo-oleanolic acid-28-oyl-hexanoate (**L12**)*


Oily solid, yield: 51.96%. ^1^H NMR (400 MHz, CDCl_3_) δ 8.11 (d, J = 7.6 Hz, 1H), 7.48 (d, J = 7.6 Hz, 1H), 5.92 (t, J = 5.3 Hz, 1H), 5.39 (s, 1H), 3.66 (s, 3H), 3.41–3.31 (m, 1H), 3.02 (dd, J = 12.9, 5.1 Hz, 1H), 2.58–2.48 (m, 1H), 1.98 (d, J = 13.5 Hz, 3H), 1.77–1.24 (m, 24H), 1.17 (s, 3H), 1.10–1.04 (m, 9H), 0.91 (s, 6H), 0.82 (s, 3H). 13C NMR (101 MHz, CDCl3) δ 217.00,174.00, 167.92,133.57, 128.46, 122.38, 77.32, 77.00, 76.68, 74.97, 71.68, 55.23, 46.79, 46.29, 42.37, 42.21, 39.22, 36.69, 34.12, 33.89, 31.90, 30.92, 30.73, 29.09, 26.42, 25.63, 24.52, 23.56, 21.47, 15.06. HRMS (ESI): [M + H]^+^ (C_37_H_59_NO_4_) calculated: 582.4881, found: 582.4894.


*3-Oxo-oleanolic acid-28-oyl-(3-((β-D-glucose)oxy)-N-(5-((β-D-glucose)oxy)p-entyl)butanamide) (**L13**)*


White solid, yield: 71.93%. ^1^H NMR (400 MHz, CDCl_3_)δ 6.72 (s, 1H), 6.12 (s, 1H), 5.38 (s, 1H), 4.29 (s, 2H), 3.84 (s, 4H), 3.55 (d, J = 35.4 Hz, 5H), 3.42–3.12 (m, 7H), 2.98 (d, J = 7.2 Hz, 1H), 2.59–2.32 (m, 4H), 1.91 (d, J = 34.7 Hz, 5H), 1.59 (s, 10H), 1.33 (d, J = 26.7 Hz, 9H), 1.15 (s, 3H), 1.07 (s, 3H), 1.03 (s, 6H), 0.88 (s, 6H), 0.78 (s, 3H). ^13^C NMR (101 MHz, CDCl_3_) δ 178.65, 178.06, 144.81, 137.52, 129.94, 111.12, 102.68, 101.32, 77.93, 77.32, 77.00, 76.68, 76.41, 75.58, 73.45, 69.75, 61.49, 55.12, 53.39, 47.37, 46.70, 46.23, 44.90, 42.07, 39.30, 39.07, 36.46, 34.06, 33.25–31.67 (m), 31.85, 31.85, 30.66, 29.64, 28.82, 27.36–27.29 (m), 26.82, 25.60, 25.22, 23.35, 22.64, 21.45, 19.27, 18.06, 17.11–16.83 (m), 16.66, 15.10. HRMS (ESI): [M + H]^+^ (C_51_H_84_N_2_O_15_) calculated: 965.5945, found: 965.5940.


*3-Oximino-oleanolic acid-28-oyl-pentanol (**L14**)*


Oily solid, yield: 81.78%. ^1^H NMR (400 MHz, CDCl_3_) δ 12.00 (s, 1H), 5.94 (s, 1H), 5.37 (s, 1H), 4.26 (s, 2H), 3.89 (d, J = 4.4 Hz, 1H), 3.63 (s, 1H), 3.34 (s, 1H), 2.51 (s, 1H), 1.89 (d, J = 14.0 Hz, 5H), 1.63 (dd, J = 32.1, 24.5 Hz, 21H), 1.13 (t, J = 9.1 Hz, 9H), 1.03 (s, 3H), 0.90 (d, J = 12.4 Hz, 9H), 0.79 (s, 3H). ^13^C NMR (101 MHz, CDCl_3_) δ 173.53, 167.91, 166.96, 140.86, 132.93, 77.32, 76.84, 76.66–76.40 (m), 69.21, 63.43, 62.30, 59.90, 55.94, 53.57, 51.66, 50.69, 49.12–42.16(m),39.59, 36.02, 32.31, 30.59,26.76, 26.67, 26.14, 23.94, 16.71, 14.72, 14.08, 8.59. HRMS (ESI): [M + H]^+^ (C_35_H_58_N_2_O_3_) calculated: 555.4520, found: 555.4520.


*3-Oximato-oleanolic acid-28-oyl-(N-(5-hydroxypentyl)hexanamide) (**L15**)*


Oily solid, yield: 54.57%. ^1^H NMR (400 MHz, CDCl_3_) δ 5.56 (s, 1H), 5.30 (s, 1H), 3.70–3.61 (m, 1H), 3.35 (tt, J = 9.8, 5.0 Hz, 4H), 3.26 (d, J = 6.4 Hz, 2H), 2.83 (s, 1H), 2.34–2.28 (m, 1H), 2.18 (t, J = 7.4 Hz, 3H), 1.87–1.71 (m, 6H), 1.60 (dtd, J = 27.0, 15.5, 7.4 Hz, 18H), 1.46–1.34 (m, 9H), 1.25 (s, 3H), 1.19–1.11 (m, 9H), 0.95–0.88 (m, 9H).^13^C NMR (101 MHz, DMSO) δ 175.57, 169.49, 164.11, 156.61, 141.10, 123.00, 50.21, 45.42, 40.20, 40.19–39.61 (m), 39.71, 39.71, 39.40, 39.08, 38.87, 36.54, 36.02, 34.24, 33.26, 32.74, 32.51, 31.97, 31.26, 31.10, 30.28, 27.48, 27.09, 25.47, 23.34, 22.94, 22.58, 21.81, 20.95, 20.78, 15.75, 14.76, 13.76, 11.18, 8.48. HRMS (ESI): [M + H]^+^ (C_41_H_69_N_3_O_4_) calculated: 668.5361, found: 668.5356.

#### 2.1.3. General Procedure to Prepare QS-21 Analogues **L16**–**L17**

Glucose trichloroacetimidate **B4** (1.15 eq) and benzyl oleanolate (1.00 eq) were dissolved in anhydrous DCM (3 mL), a 4 Å molecular sieve was added and stirred at room temperature for 40 min under **N2** conditions, TMSOTf (0.044 eq) was added dropwise, and the mixture was stirred for 20 min. After completion of the reaction, the mixture was quenched with triethylamine and filtered by diatomaceous earth extraction. The mixture was quenched with triethylamine after the reaction was completed for 20 min, filtered with diatomaceous earth, concentrated under reduced pressure, and purified by column chromatography (ethyl acetate: petroleum ether = 1:5) to obtain a white solid. The above oleanolic acid derivative (1.00 eq) was dissolved in DCM/MeOH (V:V, 1:1) (3 mL), 10% Pd/C (0.013 g) was added, and the reaction was carried out at room temperature under 1 atm H_2_ for 4 h. After the reaction, the product was purified by column chromatography (ethyl acetate: petroleum ether = 1:3) after filtration with diatomaceous earth under reduced pressure. The product was dissolved in dry DCM (3 mL), oxalyl chloride (10.13 eq) was added slowly dropwise at room temperature, it was stirred for 4 h, the reaction solution was evaporated, toluene was added and dissolved, the solution was evaporated again, and this was repeated three times to remove the solvent. The above solid product was dissolved in anhydrous DCM solution (3 mL), amine **D14**/**D24** (1.00 eq) and 5 drops of triethylamine were added, and the solution was reacted at room temperature for 4 h. After the reaction, the reaction solution was concentrated under reduced pressure, evaporated to remove the solvent, then dissolved with DCM and extracted with water several times, washed with saturated NaCl, dried with anhydrous Na_2_SO_4_, concentrated under vacuum, and purified by column chromatography. The product (1.00 eq) was dissolved in dry DCM/MeOH (1:2) (3 mL), added to freshly prepared sodium methanol–methanol (1 mol/L) solution, stirred at room temperature for 2 h, then added to acidic cation exchange resin to neutral pH = 7, extracted, concentrated, and purified by column chromatography to give target compounds **L16**, **L17**.


*3-O-(β-D-glucose)-oleanolic acid-28-oyl-(5-((β-D-glucose)oxy)pentane) (**L16**)*


White solid, yield: 71.43%. ^1^H NMR (400 MHz, MeOD) δ 5.26 (s, 1H), 4.28–4.25 (m, 2H), 3.96–3.92 (m, 2H), 3.91–3.85 (m, 4H), 3.68 (dd, J = 11.8, 5.1 Hz, 4H), 3.58 (dd, J = 8.0, 5.0 Hz, 2H), 3.36 (d, J = 4.3 Hz, 3H), 3.28 (s, 3H), 3.21–3.17 (m, 4H), 2.90–2.82 (m, 1H), 1.92 (d, J = 7.2 Hz, 2H), 1.74–1.58 (m, 17H), 1.45 (dt, J = 8.7, 6.9 Hz, 8H), 1.18 (s, 3H), 1.08 (s, 3H), 0.97 (d, J = 4.8 Hz, 6H), 0.93 (s, 3H), 0.87 (s, 3H), 0.83 (s, 3H). ^13^C NMR (101 MHz, MeOD) δ 103.96, 79.05, 78.72, 78.40, 77.71, 74.69, 74.33, 71.28, 70.30, 62.50, 49.64, 49.32, 49.00, 48.79, 48.57, 48.36, 42.59, 40.46, 40.05, 37.59, 33.50, 30.47, 30.13, 29.91, 29.20, 28.54, 28.20, 24.27, 23.95. HRMS (ESI): [M + H]^+^ (C_47_H_79_NO_13_) calculated: 866.5624, found: 866.5679.


*3-O-(β-D-glucose)-oleanolic acid-28-oyl-(N-(5-((β-D-glucose)oxy)pentyl)hex-anamide) (**L17**)*


White solid, yield: 79.19%. ^1^H NMR (400 MHz, DMSO) δ 5.20 (s, 1H), 5.04 (s, 1H), 4.14 (s, 1H), 4.09 (d, J = 7.8 Hz, 2H), 3.73 (d, J = 9.5 Hz, 1H), 3.63 (s, 2H), 3.12 (s, 1H), 3.04 (d, J = 4.7 Hz, 4H), 3.01 (s, 2H), 2.77 (d, J = 9.8 Hz, 2H), 2.01 (d, J = 4.8 Hz, 2H), 1.78 (s, 3H), 1.54–1.44 (m, 15H), 1.38–1.28 (m, 12H), 1.21 (d, J = 14.5 Hz, 4H), 1.05 (s, 3H), 0.97 (s, 6H), 0.73 (d, J = 7.5 Hz, 9H), 0.65 (s, 3H). ^13^C NMR (101 MHz, DMSO) δ 173.22, 166.54, 133.02, 131.80, 128.98, 111.76, 111.06, 102.48, 94.65, 85.54, 81.86, 76.86, 73.44, 70.13, 66.79, 65.62, 62.39, 62.04, 60.22, 59.22, 52.53, 52.15, 50.46, 49.50, 49.22–49.19 (m), 48.58, 48.00, 47.24, 46.35, 45.56, 44.72, 43.64, 43.41, 31.68, 29.90, 28.95, 27.46, 25.96, 25.12, 24.54, 24.15, 23.59, 22.97, 18.68, 17.78, 17.31, 16.90, 15.14. HRMS (ESI): [M + H]^+^ (C_53_H_90_N_2_O_14_) calculated: 979.6465, found: 979.6466.

#### 2.1.4. General Procedure to Prepare QS-21 Analogues **L18**

Raw material maltotriose trichloroacetimidate (1.15 eq) and benzyl oleanolate (1.000 g, 1.00 eq) were dissolved in anhydrous DCM (3 mL), added to a 4 Å molecular sieve, and stirred at room temperature for 40 min under **N2** conditions; TMSOTf (0.044 eq) was added dropwise, and the mixture was stirred. The product (1.00 eq) was dissolved in dry DCM/MeOH (1:2) (3 mL), freshly prepared sodium methanol–methanol (1 mol/L) solution was added, the solution was stirred at room temperature for 2 h, and then acidic cation exchange resin was added to neutralize the pH = 7. It was filtered, concentrated, and purified by column chromatography to give target compound **L18**.


*(3-O-(β-D-Maltotriose)oleanolic acid-28-yl)-benzyl ester (**L18**)*


White solid, yield: 66.56%. ^1^H NMR (400 MHz, DMSO) δ 7.38–7.24 (m, 5H), 5.69 (d, J = 4.8 Hz, 1H), 5.18 (s, 1H), 5.01 (dd, 3H), 4.92 (d, 1H), 4.14 (s, 1H), 3.86 (t, 1H), 3.60 (t, 4H), 3.49 (dd, 5H), 3.39–3.31 (m, 11H), 3.23 (s, 1H), 3.07 (d, 1H), 2.85–2.78 (m, 1H), 1.64 (s, 7H), 1.50 (dd, 13H), 1.09 (d, 3H), 0.97 (t, 2H), 0.86 (t, 12H), 0.75 (d, 1H), 0.67 (s, 3H), 0.52 (s, 3H). ^13^C NMR (101 MHz, DMSO) δ 186.87, 166.56, 136.73, 128.36, 127.79, 100.95, 94.29, 79.64, 78.84, 78.01, 76.20, 75.03, 73.51, 73.20, 72.66, 71.97–71.85 (m), 71.77, 71.85–70.80 (m), 69.93, 65.29, 60.80, 54.77, 48.58, 46.86, 46.06, 45.37, 37.95, 36.35, 32.72, 30.69, 30.37, 28.03, 27.05, 25.64, 24.68, 23.33, 22.88, 22.59, 17.99, 16.56, 15.05, 10.06. HRMS (ESI): [M + H]^+^ (C_55_H_84_O_18_) calculated: 1033.5730, found: 1033.5715.

### 2.2. Biological Evaluation

Red blood cells were obtained from normal rabbit hearts and provided by Beijing Sinovac Biotech Ltd. (Beijing, China). The single wells of a Falcon flexible round-bottom 96-well plate were numbered and 120 μL of a series of graded concentrations of PBS suspension of QS-21 and saponin analog **L1**–**L18** was added to the single wells of the well plate, repeating each set three times. We continued adding 30 μL of 10% erythrocyte suspension to the individual wells of the saponin solution and mixed gently. After incubation for 1 h in a 37 °C incubator, the supernatant was removed and centrifuged (3000 r/min, 10 min), and the absorbance value was measured at 575 nm using an enzyme marker. Four groups were established, i.e., treatment group (30 μL of prepared erythrocyte suspension plus 120 μL of QS-21 and analogues), positive control group (30 μL of prepared erythrocyte suspension plus 120 μL of Triton X-100 suspension), negative control group (30 μL of prepared erythrocyte suspension plus 120 μL of PBS/DMSO (9:1) suspension), and blank control group (30 μL of prepared erythrocyte suspension plus 120 μL of PBS/DMSO (9:1) suspension). The absorbance was measured at 575 nm in the blank control group (30 μL of saline plus 120 μL of PBS/DMSO (9:1) suspension). Each group was repeated three times. After incubation at 37 °C for 1 h according to the above procedure, we observed the color of the supernatant and the bottom of the well plate for residual erythrocytes and recorded the degree of hemolysis. After incubation at 37 °C for 1 h, the supernatant was removed and centrifuged (3000 r/min, 10 min), and the absorbance was measured at 575 nm with an enzyme marker. The hemolysis rate was calculated based on the following formula: Hemolysis rate (%) = (OD − OD _negative_)/(OD _positive_ − OD _negative_) × 100%
where OD is the absorbance of the experimental group; OD _negative_ is the absorbance of the negative control group; OD _positive_ is the absorbance of the positive control group.

## 3. Results and Discussion

### 3.1. Chemical Synthesis

Firstly, based on the structure of oleanolic acid, C-28 was amidated to improve the overall stability, and NaOMe/MeOH was used to remove the benzoyl protective group of saponins, and then the saponin analogues **L1**–**L8** of QS-21 were obtained. Then, its C3-OH was modified. One strategy was to use Dess–Martin oxidant to oxidize it into ketone, and then give an A1 unit. Another strategy was to use NH_2_OH·HCl to react with C3-oxo to give oxime and then an A2 unit. Similar amidation and benzoyl deprotection were carried out to give saponin analogues **L10**–**L15**. (Figure 1).

For the synthesis of QS-21 analogs **L16**–**L17**, glucose was used as the B fragment instead of the branched trisaccharide of QS-21, which was glycosylated with the A fragment oleanolic acid C3-OH and attached to the acyl chain D fragment, and finally the protecting group was removed to obtain the final product. (Figure 2).

Finally, QS-21 saponin analogs **L18** with maltotriose at the C3 position of oleanolic acid were synthesized with only a benzyl group attached to the C28 position of the sapogenins.

In general, four structural domains of QS-21, i.e., fragment A—triterpene unit, fragment B—trisaccharide unit, fragment C—tetrasaccharide unit, and fragment D—acyl side chain, were modified and optimized, respectively, and 18 saponin analogs were individually designed and synthesized. All the synthesized compounds gave satisfactory analytical and spectroscopic data in full agreement with their described structures. 

### 3.2. Biological Evaluation

The hemolysis rates of those synthesized QS-21 analogues (**L1**–**L18**) were assayed with the commercial adjuvant QS-21 as control (Table 1). The hemolysis rates are shown in Table 1.

Generally, compared with QS-21, most of the QS-21 analogs showed lower hemolysis rates at a concentration of 500 μg/mL, and low hemolysis effects were observed in all samples, while the relatively higher hemolysis rates of **L12** and **L15** indicated that they were more likely to cause erythrocyte disintegration and induce hemolytic effects.

The SAR of these QS-21 analogues was initially summarized as follows. (1) **L1**–**L8**, which have no modification at C3, produced little hemolysis, indicating that for the unmodified saponins at the C3 position of the glycoside, there was no significant difference in the effect of different lengths of sugar chains attached to the ends of the acyl side chains on their hemolysis. (2) Compared with **L2**–**L3**, **L5** and **L7** with C3-OH and **L10**–**L12** with 3-oxo showed stronger hemolytic effects at different concentrations, which indicated that 3-oxo can enhance the overall hemolytic effects. For the 3-oxime analogues, **L15** showed higher hemolytic effects than that of **L14** at different concentration gradients, perhaps because of its high hydrophobicity. (3) **L16**–**L18,** whose glycosides were attached to the C3 position, produced no hemolysis, indicating that sugar molecules attached to the C3 position can reduce hemolytic effects, while the excision of the acyl side chain at the C28 position did not increase its hemolytic effects. All the NMR and HRMS spectra of synthesized compounds **L1**–**L18** are provided in Appendix A.

## 4. Conclusions

Although QS-21 has powerful immunostimulatory effects, only two vaccines, the Herpes Zoster vaccine (Shingrix^TM^(GSK, London, UK)) and RTS,S/AS01 vaccine (Mosquirix^TM^(GSK, London, UK)), containing QS-21 have been approved for marketing until now [1]. Its structural complexity, scarcity, chemical instability, severe immune side effects, expensive and restricted intellectual property procurement, dose-limiting toxicity, and adverse hemolytic effects have limited its further application. In the study, a series of novel QS-21 analogs was designed and synthesized and a biological evaluation of their hemolytic properties was performed. Lower hemolysis effects were observed in most of the analogs at a concentration of 500 μg/mL than that of QS-21, and they are expected to serve as potent immunogenicity to vaccine formulations in the future. It will be necessary to conduct more comprehensive immunopotentiation studies and perform the toxicological evaluation of them to lay the foundation for the development of a new QS-21 immunoadjuvant analogue with low toxicity and high efficiency.

## Data Availability

The data presented in this study are available in this article and the Appendix A.

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
