# Peer review of "Design and Synthesis of Immunoadjuvant QS-21 Analogs and Their Biological Evaluation"

_biomedicines, 2024, doi:10.3390/biomedicines12020469_

Round 1
Reviewer 1 Report
Comments and Suggestions for Authors
Title: “Design and synthesis of immunoadjuvant QS-21 analogues and their biological evaluation”
In this research article, the authors have presented a study on a series of novel immunoadjuvant QS-21 analogues, evaluating their effects on in vitro hemolysis of red blood cells, using QS-21 as a control and hemolytic properties as an index. The results indicate that the QS-21 analogues exhibit lower hemolytic side effects compared to QS-21, establishing a quantitative-effect relationship between their concentrations and the hemolysis rate. Saponin compounds L1-L9, in particular, demonstrate minimal hemolysis, suggesting their potential as immunoadjuvants with reduced hemolytic side effects.
The findings suggest lower hemolysis rates for these analogues, especially compounds L1-L9, contribute to the potential safety improvement of immunoadjuvants. The manuscript is well-organized and written with clarity, making it accessible to both experts and non-experts in the field.
While the manuscript effectively presents the observed hemolytic effects and the quantitative relationship, it would be beneficial to include discussions or speculations on the potential mechanisms behind the reduced hemolytic side effects. To strengthen the manuscript, it would be valuable to include a more detailed comparative analysis with existing literature on immunoadjuvants. Authors should provide the full spectral data in supporting information. Also, the authors should check and correct the English language ad correct the grammar throughout the manuscript. It could be accepted for the publication in “Biomedicines” after minor revision.

The authors are advised to carefully review and rectify the English language, ensuring proper grammar throughout the manuscript.
Author Response
Comments 1:
Title:“Design and synthesis of immunoadjuvant QS-21 analogues and their biological evaluation”
In this research article, the authors have presented a study on a series of novel immunoadjuvant QS-21 analogues, evaluating their effects on in vitro hemolysis of red blood cells, using QS-21 as a control and hemolytic properties as an index. The results indicate that the QS-21 analogues exhibit lower hemolytic side effects compared to QS-21, establishing a quantitative-effect relationship between their concentrations and the hemolysis rate. Saponin compounds L1-L9, in particular, demonstrate minimal hemolysis, suggesting their potential as immunoadjuvants with reduced hemolytic side effects.
The findings suggest lower hemolysis rates for these analogues, especially compounds L1-L9, contribute to the potential safety improvement of immunoadjuvants. The manuscript is well-organized and written with clarity, making it accessible to both experts and non-experts in the field.
While the manuscript effectively presents the observed hemolytic effects and the quantitative relationship, it would be beneficial to include discussions or speculations on the potential mechanisms behind the reduced hemolytic side effects. To strengthen the manuscript, it would be valuable to include a more detailed comparative analysis with existing literature on immunoadjuvants. Authors should provide the full spectral data in supporting information. Also, the authors should check and correct the English language ad correct the grammar throughout the manuscript. It could be accepted for the publication in “Biomedicines” after minor revision.
Response 1:
Thank you very much for your valuable comments on the manuscript. Based on your comments, we have made the following reply and modification:
(1) As you pointed out that it is really necessary to speculate and explore the potential mechanism of reducing the hemolytic effects. In the third part of the manuscript, we have a series of discussions on the hemolysis rate results of the synthesized QS-21 analogues based on the compound structure, combined with the biological evaluation experimental results.
(2) The introduction section (Page 2, Lines 44-54) in the revised manuscript has been supplemented.
(3) All the spectral data information was listed in the third part of the manuscript.
(4) We have checked and corrected the English language ad corrected the grammar throughout the manuscript.

Reviewer 2 Report
Comments and Suggestions for Authors
The manuscript described the synthesis and hemolysis rate of 20 new compounds. The most important problem is wrong construction of manuscript. The materials and method contain to many information, while results and discussion sections are too short. I recommended in material and method parts add only the methodology of synthesis and biological assay. In results section could be divided into two parts, in first describes the synthesis and in second describes the biological assay. Discussion should summarize the results. Authors should added to supplementary materials 1H, 13C and MS spectra for all compounds.
Manuscript contain minor editional problem, like to low Figures 1 and 2, or 3.000 mL should be change to 3 mL.
Author Response
Comments 2:
The manuscript described the synthesis and hemolysis rate of 20 new compounds. The most important problem is wrong construction of manuscript. The materials and method contain to many information, while results and discussion sections are too short. I recommended in material and method parts add only the methodology of synthesis and biological assay. In results section could be divided into two parts, in first describes the synthesis and in second describes the biological assay. Discussion should summarize the results. Authors should added to supplementary materials 1H, 13C and MS spectra for all compounds.
Manuscript contain minor editional problem, like to low Figures 1 and 2, or 3.000 mL should be change to 3 mL.
Response 2:
Thank you very much for your valuable comments on the manuscript. Based on your comments, we have made the following reply and modification:
(1) Due to the improper arrangement of the structure of the manuscript, the relevant spectral data of all the synthesized compounds are listed in the materials and methods section, which makes this part of the content too complicated. In the present manuscript, the section order has been changed .
(2)The 1H, 13C and MS spectra of all compounds L1-L20 synthesized in this work have been listed in the supplementary material.
(3) Minor editional problems in the manuscript you mentioned have been corrected.

Reviewer 3 Report
Comments and Suggestions for Authors
Page 1: line19, defined “Th1 or Th2” when first appears.
Be careful of the conclusion, which should only derive from the data. Please re-analyze the SAR carefully.
Page 10.
“L11- L14 with 3-oxo or 3-oxime showed strong hemolytic effects at different concentrations, which indicated that 3-oxo saponins are more likely to produce 3erythrocyte hemolytic side effects; the results of the hemolysis rate of L11, L12 showed 381 that for the 3-oxo saponin, monosaccharide or disaccharide at the end of the side chain can increase the hemolysis;” This statement regarding SAR may be incorrect because L11-L15 share the same 3-oxo group and the only difference lies in R2 group, L15 has low hemolysis and the degree of hemolysis between L11 to L14 are different too, though L11-L15 share the same 3-oxo group. Thus, the difference can only come from the R2 group, rather than the 3-oxo group.
“for the 3-oxime saponin L17 385 produced strong hemolysis at different concentration gradients, probably because the C3-oxime functional group tends to promote the interaction between saponin and cholesterol 387 on the surface of the red blood cell wall.” This statement regarding SAR may be incorrect because L16 contains C3-oxime group like L17, but L16 has low hemolysis. The difference between L16 and L17 lies in the extra amide group with extra 5 carbon extension. Thus the contribution may come from hydrophobicity, not the oxime group.
The references need to update. Many references are quite old and the latest one is 2019. There are quite a few applications of QS21 published in recent years, like in 2023, even in January 2024. The abbreviation of journal names need to be checked for correctness. For instance, the following errors in the references should be correct.
Page 11, “Phytomedicin “ should be “Phytomedicine “
Page 11, “Chemical Science ” should be “Chem Sci ”
Page 11, “in African children[J]. ” should be “in African children. ”
Page 12, “Human vaccines ” should be “Hum vaccin ”
Author Response
Comments 3:
Page 1: line19, defined “Th1 or Th2” when first appears.
Be careful of the conclusion, which should only derive from the data. Please re-analyze the SAR carefully.
Page 10.
“L11-L14 with 3-oxo or 3-oxime showed strong hemolytic effects at different concentrations, which indicated that 3-oxo saponins are more likely to produce 3erythrocyte hemolytic side effects; the results of the hemolysis rate of L11, L12 showed 381 that for the 3-oxo saponin, monosaccharide or disaccharide at the end of the side chain can increase the hemolysis;” This statement regarding SAR may be incorrect because L11-L15 share the same 3-oxo group and the only difference lies in R2 group, L15 has low hemolysis and the degree of hemolysis between L11 to L14 are different too, though L11-L15 share the same 3-oxo group. Thus, the difference can only come from the R2 group, rather than the 3-oxo group.
“for the 3-oxime saponin L17 385 produced strong hemolysis at different concentration gradients, probably because the C3-oxime functional group tends to promote the interaction between saponin and cholesterol 387 on the surface of the red blood cell wall.” This statement regarding SAR may be incorrect because L16 contains C3-oxime group like L17, but L16 has low hemolysis. The difference between L16 and L17 lies in the extra amide group with extra 5 carbon extension. Thus the contribution may come from hydrophobicity, not the oxime group.
The references need to update. Many references are quite old and the latest one is 2019. There are quite a few applications of QS21 published in recent years, like in 2023, even in January 2024. The abbreviation of journal names need to be checked for correctness. For instance, the following errors in the references should be correct.
Page 11,“Phytomedicin “should be “Phytomedicine “
Page 11,“Chemical Science” should be “Chem Sci”
Page 11,“in African children[J].” should be “in African children. ”
Page 12,“Human vaccines” should be “Hum vaccin”
Response 3:
(1) We have defined “Th1 or Th2” when first appears.(Page 1, Line 19)
(2) The previous analysis based on the results of biological evaluation experiments is not accurate and rigorous. Now, a further discussion has been made .
(3) We have checked and revised the abbreviations of journal names that appear in the references, and several errors you mentioned have been corrected.

Reviewer 4 Report
Comments and Suggestions for Authors
The authors present some interesting approach to creating a family of synthetic adjuvants based on the saponin QS-21. They have considered the molecule as have four structural segments and then vary these segments synthetically substituting different sugars (more common ones than in natural QS-21) and varying functional groups. The family of 20 compounds are tested as to their tendency to lyse red blood cells. The potentially exciting result is that some of the derivatives show much less or minimal hemolysis compared to QS-21. Their immune response enhancement as adjuvants will presumably now be the subject of follow up major study. The paper is important, and a few improvements can be made:
(1) Please define Th1 and Th2 when they first appear
(2) line 372, these were not really patients and should be called samples.
(3) The conclusion can be enhanced by briefly describing how these compounds will be studied in the future with respect to their potential utility in vaccines. Is it animal models? What is a good antigen to test?
(4) lines 39-40 should add some references.
Comments on the Quality of English LanguageThe English usage seems good.
Author Response
Comments 4:
The authors present some interesting approach to creating a family of synthetic adjuvants based on the saponin QS-21. They have considered the molecule as have four structural segments and then vary these segments synthetically substituting different sugars (more common ones than in natural QS-21) and varying functional groups. The family of 20 compounds are tested as to their tendency to lyse red blood cells. The potentially exciting result is that some of the derivatives show much less or minimal hemolysis compared to QS-21. Their immune response enhancement as adjuvants will presumably now be the subject of follow up major study. The paper is important, and a few improvements can be made:
(1) Please define Th1 and Th2 when they first appear
(2) line 372, these were not really patients and should be called samples.
(3) The conclusion can be enhanced by briefly describing how these compounds will be studied in the future with respect to their potential utility in vaccines. Is it animal models? What is a good antigen to test?
(4) lines 39-40 should add some references.
Response 4:
Thank you very much for your valuable comments on the manuscript. Based on your comments, we have made the following reply and modification:
(1) We have defined Th1 and Th2 when they first appear.
(2) We have changed "patients" to "samples" on line 372.
(3) Thanks for your valuable advice. We have improved the summary part according to your suggestion.
(4) For the content of lines 39-40, we have added three relevant references and added some discussion to their work.(Pgae 2, Lines 44-54)

Round 2
Reviewer 2 Report
Comments and Suggestions for Authors
The manuscript is poorly corrected. Now it has many more bugs than the first version. I suggest you read the review carefully. In addition, I recommend reading other articles on the synthesis and evaluation of biological activity so that the authors can see what a scientific article should look like.
The 1H and 13C spectra for many compounds are of very poor quality. For example, the 1H NMR spectrum for L6 contains too few signals. In the 13C spectra, the signal mixes with the noise line.
Author Response
Comments 2:
The manuscript is poorly corrected. Now it has many more bugs than the first version. I suggest you read the review carefully. In addition, I recommend reading other articles on the synthesis and evaluation of biological activity so that the authors can see what a scientific article should look like.
The 1H and 13C spectra for many compounds are of very poor quality. For example, the 1H NMR spectrum for L6 contains too few signals. In the 13C spectra, the signal mixes with the noise line.
Response 2:
(1) Thank you for your valuable comments and suggestions, and I am very sorry for not being able to solve the problem you raised last time. These comments are very helpful for the revision and improvement of our paper. We have carefully studied these comments and revised our manuscript format with reference to two articles in Biomedicines (listed below). Firstly, in the part of materials and methods, only the methods of chemical synthesis and bioassay are described, and the 1H, 13C and MS spectrogram data of 20 QS-21 analogues designed and synthesized are listed. Results and discussion are divided into two parts, the first part describes chemical synthesis, the second part describes biological assay, and summarizes the conclusions.
[1] Brito V, Marques M. Synthesis, In Vitro Biological Evaluation of Antiproliferative and Neuroprotective Effects and In Silico Studies of Novel 16 E-Arylidene-5α, 6α-epoxyepiandrosterone Derivatives. Biomedicines, 2023, 11: 812. https://pubmed.ncbi.nlm.nih.gov/36979790/
[2] Sowa-Kasprzak K, Totoń E. Synthesis, Cytotoxicity and Molecular Docking of New Hybrid Compounds by Combination of Curcumin with Oleanolic Acid. Biomedicines, 2023, 11: 1506. https://pubmed.ncbi.nlm.nih.gov/37371601/
(2) We have tried our best to purify the target sugar compounds to improve the quality of their corresponding 1H and 13C spectra through column separation and crystallization. However, as you can see, the result is not perfect. Other researchers have also found that it is very difficult to obtain high-quality spectra of QS-21 analogues(J Org Chem. 2016, 81: 9560-9566.). There is also a research topic in our future research.
[1] Wang P, Devalankar DA. Synthesis and Evaluation of QS-21-Based Immunoadjuvants with a Terminal-Functionalized Side Chain Incorporated in the West Wing Trisaccharide. J Org Chem. 2016, 81: 9560-9566. https://pubmed.ncbi.nlm.nih.gov/27709937/

Reviewer 3 Report
Comments and Suggestions for Authors
The authors have addressed most of the main issue. Though the authors stated that they had defined "Th1 or Th2", but they did not. Other than the font became red, nothing was changed.
Page 1: line19, defined “Th1 or Th2” when first appears.
By looking carefully at Figure 2, the exo ring structure of C=O (A1) and C=N-OH was not 120 degrees for the sp2 carbons. It looks very odd. Please redraw A1 and A2 with C=O or C=N double bond so that the sp2 C should be 120 degrees.
These are just minor changes and should not require another round of peer review.
Author Response
Comments 3:
The authors have addressed most of the main issue. Though the authors stated that they had defined "Th1 or Th2", but they did not. Other than the font became red, nothing was changed.
Page 1: line19, defined “Th1 or Th2” when first appears.
By looking carefully at Figure 2, the exo ring structure of C=O (A1) and C=N-OH was not 120 degrees for the sp2 carbons. It looks very odd. Please redraw A1 and A2 with C=O or C=N double bond so that the sp2 C should be 120 degrees.
These are just minor changes and should not require another round of peer review.
Response 3:
Thanks for your careful checks. We are sorry for our carelessness.
(1) Based on yourcomments, we have defined “Th1 or Th2” when first appears.(Page 1: line 21)
(2) We have redrawn A1 and A2 with C=O or C=N double bond.(Figure 2,Page 3)

Round 3
Reviewer 2 Report
Comments and Suggestions for Authors
The manuscript is better but still needs corrections. The Authors should read all manuscript and correct all mistake. For example, in Line 354 is “Results and Discussion” but line 395 “Discussion”. The 13C NMR spectra are poor quality, the size of signals and noise are similar, e.g. Figure S8 a signal at a value of 157.95 ppm is a signal and a signal of the same height at a value of approximately 175 ppm is considered noise. 13C NMR spectra should be performed again, extending the spectrum recording time.
Author Response
Comments 2:
The manuscript is better but still needs corrections. The Authors should read all manuscript and correct all mistake. For example, in Line 354 is “Results and Discussion” but line 395 “Discussion”. The 13C NMR spectra are poor quality, the size of signals and noise are similar, e.g. Figure S8 a signal at a value of 157.95 ppm is a signal and a signal of the same height at a value of approximately 175 ppm is considered noise. 13C NMR spectra should be performed again, extending the spectrum recording time..
Response 2:
Thank you very much for your time and valuable suggestions on our revised manuscript.
(1) Corrections have been made to address inconsistencies before and after headings in lines 354 and 395, and the manuscript has been meticulously examined.
(2) Due to the large molecular weight and low concentration of some samples, in order to get better quality of spectra, sufficient scanning time was used during the 13C NMR test for each sample, and some samples were even scanned overnight. Regarding the 13C NMR test of L3, we tried to extend the scanning period, however the spectrum did not significantly improved. As mentioned in the previous reply, other researchers have also found that it is very difficult to obtain high-quality spectra of QS-21 analogues which means that it is a common problem in the synthesis of QS-21 compounds. This also will be our key research topic in the future.

Round 4
Reviewer 2 Report
Comments and Suggestions for Authors
High molecular weight is not a problem in recording the 13C NMR spectrum. The presented spectra do not confirm the structure of the compounds. Moreover, HRMAS are of questionable quality. HRMAS spectra should contain only the molecular peak and possible fragmentary peaks. Authors must add the total formula for the compound because it is difficult to check whether the mass of the compound has been reported correctly.
Author Response
Comments 2:
High molecular weight is not a problem in recording the 13C NMR spectrum. The presented spectra do not confirm the structure of the compounds. Moreover, HRMAS are of questionable quality. HRMAS spectra should contain only the molecular peak and possible fragmentary peaks. Authors must add the total formula for the compound because it is difficult to check whether the mass of the compound has been reported correctly.
Response 2:
Thank you very much for your time and valuable suggestions on our revised manuscript.
Regarding the problem with the HRMAS spectra you pointed out, we have checked the manuscript and supplementary materials and found some problems with the spectra of compounds L9 and L14. However, due to the Spring Festival holiday, it is inconvenient for us to conduct supplementary experiments, so we chose to delete the relevant data of compounds L9 and L14 to ensure the accuracy of the manuscript, and we have added the total formula for the compound in the manuscript. Thank you again for your valuable comments on our manuscript.
